# RiDOP: A Rotation-Invariant Detector with Simple Oriented Proposals in Remote Sensing Images

**Chongyang Wei** [1,*]**, Weiping Ni** [1]**, Yao Qin** [1]**, Junzheng Wu** [1,2]**, Han Zhang** [1,2]**, Qiang Liu** [1,3]**, Kenan Cheng** [1]
**and Hui Bian** [1]

1    Northwest Institute of Nuclear Technology, Xi'an 710024, China
2    College of Electronic Science, National University of Defense Technology, Changsha 410073, China
3    School of Software Engineering, Xi'an Jiaotong University, Xi'an 710049, China
*    Correspondence: weichongyang@nint.ac.cn

**Abstract:** Compared with general object detection with horizontal bounding boxes in natural images, oriented object detection in remote sensing images is an active and challenging research topic as objects are usually displayed in arbitrary orientations. To model the variant orientations of oriented objects, general CNN-based methods usually adopt more parameters or well-designed modules, which are often complex and inefficient. To address this issue, the detector requires two key components to deal with: (i) generating oriented proposals in a light-weight network to achieve effective representation of arbitrarily oriented objects; (ii) extracting the rotation-invariant feature map in both spatial and orientation dimensions. In this paper, we propose a novel, lightweight rotated region proposal network to produce arbitrary-oriented proposals by sliding two vertexes only on adjacent sides and adopt a simple yet effective representation to describe oriented objects. This may decrease the complexity of modeling orientation information. Meanwhile, we adopt the rotation-equivariant backbone to generate the feature map with explicit orientation channel information and utilize the spatial and orientation modules to obtain completely rotation-invariant features in both dimensions. Without tricks, extensive experiments performed on three challenging datasets DOTA-v1.0, DOTA-v1.5 and HRSC2016 demonstrate that our proposed method can reach state-of-the-art accuracy while reducing the model size by 40% in comparison with the previous best method.

**Keywords:** oriented object detection; object representation; rotation-invariant feature; image processing; remote sensing





## 1. Introduction

Object detection has gained promising progress with the rapid development of convolutional neural network (CNN) in the past decade. Different from objects in natural images, objects of remote sensing images are usually distributed with arbitrary orientation and densely packed. Therefore, the detection in remote sensing images is generally formulated as the oriented object detection. The traditional detection methods in natural images, often routinely locating the objects with Horizontal Bounding Boxes (HBBs), are obviously inappropriate for oriented objects because the Regions of Interest (RoI) of HBBs are easily contaminated with extra pixels from background or nearly interleaved other objects. In contrast, the oriented object detection methods [1–3] adopt Oriented Bounding Boxes (OBBs) to locate the objects instead of using HBBs [4–6]. However, general CNN-based detection methods usually adopt more parameters or well-designed modules to model the arbitrary orientation information of oriented objects, which are highly redundant and inefficient. Moreover, a large amount of rotation augmented data is necessary to train an accurate object detector. To resolve this issue, this detection task in unconstrained remote sensing images has two key components to deal with: (i) generating high-quality oriented proposals in the manner of low computational complexity and using them to represent

efficiently arbitrarily oriented objects; (ii) extracting rotation-invariant features to make sure that the features do not change with orientations of objects.

There are three main categories of generating oriented proposals to encode orientation information. The early category is to use rotated Region Proposal Network (RPN) [7–9] to encode different angles, aspect rations and scales on each location of all levels in the feature map. It generates a large number of anchors (the number is equal to num_angles × num_aspect_ratios × num_scales × num_locations in each level × num_feature_levels) in the total feature map. The introduction of rotated anchors improves the recall of detection and proves to have promising potential in locating sparsely distributed arbitrary-oriented objects. However, owing to the highly various orientations of objects in remote sensing images, it is necessary to more finely encode the oriented angle to reach greater recall in the detection task, which results in occupying massive computational resources. Another category adopts a conventional network to learn the transformation [5] from horizontal RoIs to oriented proposals by a complex process, which consists of generating horizontal RoIs with RPN, aligning ones with ground truth boxes annotated by human and regressing rotated RoIs. It generates promising rotated proposals and considerably decreases the number of encoded rotated anchors, but also results in the expensive computational cost. The last category is to glide four vertexes of the HBB on the corresponding sides to represent a rotated object and adopt an obliquity variable $r$ to characterize the tilt degree of the rotated object [10]. This method employs horizontal RoIs to perform classification and rotated proposals to execute regression, which will result in severe misalignment between oriented objects and horizontal features. Consequently, it is necessary to design an efficient and lightweight solution to generate orientated proposals.

Most state-of-the-art oriented object detectors are devoted to extracting rotation-invariant features [5,11,12] to keep the features stable and do not change with the orientations of rotated objects. Specifically, Rotated RoI (RRoI) Pooling [9] and RRoI Align [5,13] are the most commonly adopted modules for this task. The modules can warp oriented region features accurately according to the bounding boxes of RRoIs in the two-Dimensional (2D) planar. However, RRoI warping with regular CNN features cannot generate rotation-invariant features in the orientation dimension [14]. The classical rotation invariance is approximated by adopting more parameters in larger capacity models and more training samples to encode the orientation information of object level [15]. This manner of encoding is usually unstable and delicate following the variety of orientation information. Recently, some promising methods apply group convolutions [16] to the input to produce transformation of the output feature in a predictable fashion. The generated feature maps use additional channels to record information from different orientations, then, the general RRoI warping is directly adopted to the feature maps. Unfortunately, it only produces rotation-invariant features in the spatial dimension of 2D planar, leaving the orientation channel misaligned. To resolve this problem, it is necessary to adopt a novel module to achieve completely rotation-invariant feature.

In this paper, we propose to combine a novel oriented RPN with rotation-invariant feature alignment module to detect arbitrary oriented objects in remote sensing images, namely RiDOP. As shown in Figure 1, our RiDOP consists of a rotation-equivariant backbone, an efficient oriented RPN , a rotation-invariant RoI (RiRoI) alignment and a detection head. Given an input image, firstly, we adopt the backbone with Rotation-equivariant ResNet (ReResNet) and Rotation-equivariant Feature Pyramid Network (ReFPN) to extract feature with explicit orientation channel information. To reduce the computational complexity of modeling oriented proposals, we propose a lightweight oriented RPN by sliding two vertexes on adjacent sides of the external rectangle of the HBB and employ a simple representation to describe oriented objects. Then, we utilize the spatial alignment module to align region features according to the bounding boxes of RRoIs and employ orientation alignment module to switch circularly the orientation channel for obtaining completely rotation-invariant features in both two dimensions. Finally, the features are fed into the detection head to classify objects and regress accurate positions.

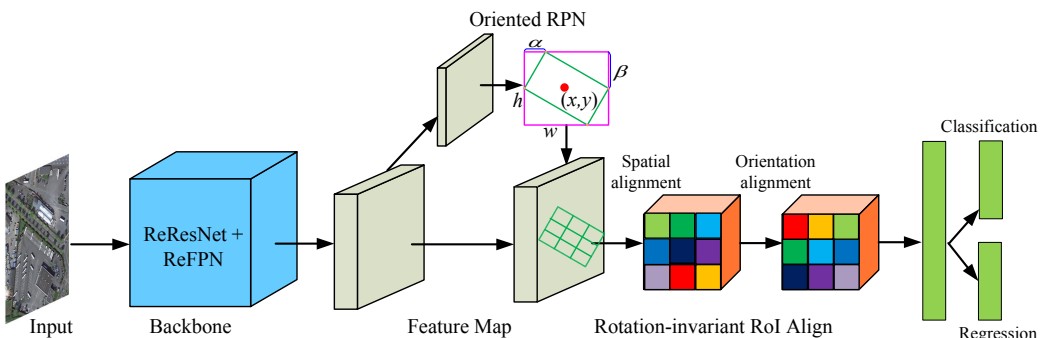

**Figure 1.** The overall architecture of our proposed method. For an input image, we first incorporate rotation-equivariant ResNet with rotation-equivariant FPN into the backbone to extract the feature, and then use lightweight oriented RPN by gliding two offset on adjacent sides to generate RRoIs, later, we adopt the rotation-invariant RoI module to align features in both spatial and orientation dimensions, finally, the detection head (classification and regression branches) are applied to produce the final detections. In the orientation alignment module, the feature map of RRoI with the shape of ($K \times K \times C$) is circularly converted to the first orientation channel (red channel). (Best viewed in color.)

Extensive experiments performed on challenging benchmarks for oriented object detection in remote sensing images, namely DOTA [1] and HRSC2016 [17], demonstrate that our proposed method can achieve state-of-the-art performance. In summary, the main contributions of our paper are as follows: (i) We propose a novel oriented RPN to generate high-quality oriented proposals and utilize them to represent oriented objects effectively, which avoids massive number of RRoIs designed for modeling orientation information and thus may decrease the number of parameters. (ii) We adopt the rotation-equivariant backbone to generate the feature map, which explicitly encodes discrete orientation channel information. Moreover, we adopt the spatial and orientation modules to align features to obtain completely rotation-invariant features in both two dimensions, which ensures that the feature is stable and robust. (iii) Without tricks, our RiDOP achieved the state-of-the art accuracy with 67.99% mAP (increased by 1.13%) on the more challenging DOTA-v1.5 and 98.47% mAP (increased by 0.84%) on HRSC2016 while reducing the number of parameters by 40% (121 Mb versus 71 Mb) in comparison with the previous best result. The experimental results demonstrated the effectiveness and superiority of our proposed method.

## 2. Related Works

With the fast development of CNN in the past decade, the general horizontal object detection methods have made promising progresses [18–27]. As objects in remote sensing images are usually distributed with arbitrary orientation and densely packed, the objects of directly using HBBs may contain more extra pixels from background or nearly interleaved other objects, which brings about a problem of misalignment between horizontal region feature and rotated object instance, and finally leads to the inconsistency between high classification score and low localization accuracy. Based on this, the object detection in remote sensing images is generally formulated as the oriented detection task.

Many well-designed methods are devoted to detect oriented objects using OBBs for obtaining accurate orientation information [4–6,13,28–30]. The early ones adopted considerable rotated anchors with different angles, aspect rations and scales on each location of feature map [7,9], which achieved promising recall in detecting sparsely distributed oriented objects but occupied massive computational resources. Yang et al. [6] introduced the denoising of instance-level on the feature map to improve the detection to cluttered small objects, and meanwhile, he also incorporated a scale factor into the smooth L1 loss function to solve the unexpected increase of loss caused by periodic angular variation and the switch between long and short edges in the boundary condition. Yang et al. [31] further proposed to adopt a 2-D Gaussian distribution to represent an OBB, and then

approximately calculate the regression loss based on the Kullback–Leibler Divergence (KLD) entropy to overcome the serious accuracy drop caused by a slight angle error for large-aspect-ratio objects. It was shown that KLD entropy could adjust dynamically the gradient weight of the angle parameter following the variety of the aspect ratio and achieved high-precision detection. In [32], a novel pixels-IoU loss function was proposed to reduce the overall IoU for dealing with the insensitive of classical angle error to rotated objects with large aspect ratios. Ding et al. [5] adopted an RRoI learner to encode horizontal RoIs and utilized position-sensitive RRoI module to decode rotated transformation, which significantly reduced the amount of generated anchors. Xu et al. [10] adopted a quadrangle by gliding four vertexes of the HBB on their corresponding sides to represent a rotated object. In [11], S$^2$A-Net was proposed to align the feature between horizontal receptive fields and rotated anchors by the feature alignment and oriented detection modules. In [29], Pan et al. adopted feature selection to dynamically modify receptive fields following the variety of orientations and shapes of target, and meanwhile, used refinement head network to dynamically refine the oriented object. In [33], Ming et al. adopted an anchor learning strategy to estimate the localization confidence and assigned an efficient label to high-quality anchors. To deal with the problem of discontinuity boundary, in [34], the detection task was formulated as coded labels problem for classifying angle densely, which improved the performance of detecting objects similar to squares.

The classical CNN networks have good properties for the generalization of translation-invariant features while show unsatisfied performance on rotation-invariant features. Cohen et al. [35] first introduced Group equivariant CNN (G-CNNs) into regular CNNs to reduce sample complexity for obtaining rotation-invariant feature. Compared with regular convolution layers, G-CNNs used group convolution layers to share model's weight with a substantially higher degree. Zhou et al. [36] proposed Active Rotating Filters (ARFs) which actively rotated during convolution by interpolation and produced the feature maps with explicit orientation channel and location information. Using ARFs could create within-class rotation-invariant feature while maintaining inter-class discrimination only for classification task. Cheng et al. [37] incorporated a regularization constraint into the optimization of the objective function in the rotation-invariant layer. In [38], a novel deformable convolution layer was created by replacing the regular layer to aggregate multi-layer features. Ding et al. [5] adopted the rotated position sensitive RoI align to warp region features to obtain the rotation-invariant feature only in the spatial dimension. Lu et al. [16] applied sparse group convolutions to the input transformations to produce transformation of the output feature in a predictable fashion, the generated feature map used additional channels to record information from different orientations. Then, the general RRoI warping was directly adopted to the feature map. Unfortunately, it only produced rotation-invariant features in the spatial dimension in 2D planar, leaving the orientation channel misaligned.

In contrast with the above methods, our method focuses on: (i) designing a lightweight oriented RPN to generate high-quality oriented proposals with a low computational budget and use those proposals to effectively represent rotated objects; (ii) extracting rotation-invariant feature with the explicit spatial and orientation channel information. Specifically, our RiDOP generates a rotation-equivariant feature map with the backbone, and then the oriented RPN is adopted to generate rotated RoIs to significantly reduce the computational complexity of modeling orientation information, finally, the RiRoI Align extracts rotation-invariant feature in the spatial and orientation dimensions to accurately locate oriented objects, as shown in Figure 1.

## 3. Proposed Method

As illustrated in Figure 1, the proposed method consists of a backbone with Rotation-equivariant ResNet (ReResNet) and Rotation-equivariant FPN (ReFPN) based on e2cnn [14], a simple yet effective oriented RPN, a rotation-invariant RoI align (RiRoI Align) module and detection head network. Given an input image, the ReResNet with ReFPN backbone is devoted to acquire a rotation-equivariant feature map, the oriented RPN is adopted to

generate rotated RoIs by sliding two vertexes on adjacent sides of the external rectangle of HBBs. With RRoIs in the feature map, RiRoI Align first adopts spatial alignment to obtain a rotation-invariant feature in the spatial dimension and then employs orientation alignment to gain rotation-invariant feature in the orientation dimension. The feature is finally fed into the classification and regression branches to produce final detections.

### 3.1. Oriented Proposal Generation

After extracting the feature from an input image with backbone, a sparse set of oriented proposals with objectness scores is generated with a lightweight fully convolutional oriented RPN network.

Specifically, this small network takes five input levels of feature maps $\{F_2, F_3, F_4, F_5, F_6\}$ from ReFPN as input. The feature of each level is fed into a $3 \times 3$ sliding spatial window, followed by two $1 \times 1$ sibling convolutional layers—one for regressing the locations of boxes and the other for classifying boxes. At each location of the sliding window, we sample $3 \times 5$ horizontal anchors with 3 aspect ratios {1:2, 1:1, 2:1} and 5 scales in all levels of features $\{F_2, F_3, F_4, F_5, F_6\}$. As illustrated in Figure 2a, each horizontal anchor $\gamma$ is represented with a 4-dimensional vector $\gamma = (\gamma_x, \gamma_y, \gamma_h, \gamma_w)$, where $(\gamma_x, \gamma_y)$ denotes the center coordinate of the anchor, $\gamma_h$ and $\gamma_w$ express the height and width of the anchor, respectively. The box-regression layer generates the offset $\Delta = (\Delta_x, \Delta_y, \Delta_h, \Delta_w, \Delta_\mu, \Delta_v)$ of the oriented proposals relative to the horizontal anchors. The oriented proposals can be obtained by decoding the output of the box-regression layer. The process can be depicted as follows:

$$\begin{cases} x = \Delta_x \cdot \gamma_w + \gamma_x, & y = \Delta_y \cdot \gamma_h + \gamma_y \\ w = \gamma_w \cdot e^{\Delta_w}, & h = \gamma_h \cdot e^{\Delta_h} \\ \mu = \Delta_\mu \cdot w, & v = \Delta_v \cdot h \end{cases} \tag{1}$$

where $(x, y)$, $h$ and $w$ are the coordinate of the center, height and width of the externa bounding box of the predicted rotated proposal, respectively. $\mu, v$ are the offsets relative to the top-left vertex and top-right vertex of the external bounding box, respectively. Finally, we generate oriented proposals by a 6-dimensional vector $(x, y, w, h, \mu, v)$. The box-classification layer is applied to estimates the confidence score of each predicted proposal.

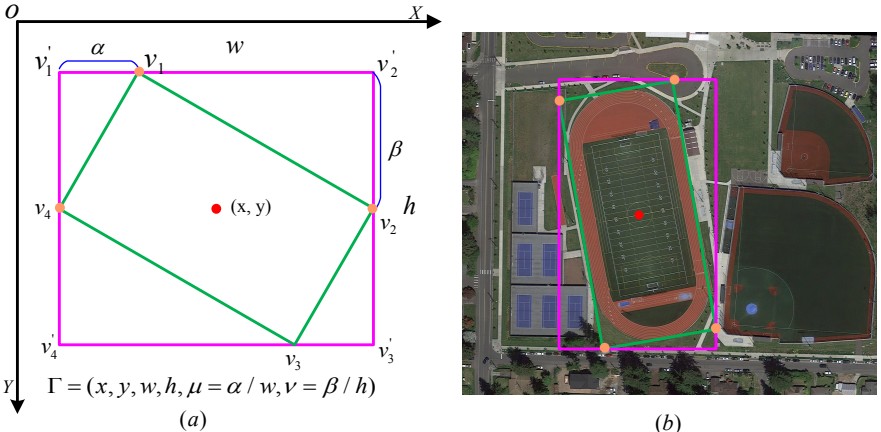

**Figure 2.** (**a**) The schematic illustration of oriented object representation with 6 parameters $\Gamma$ {$x, y, w,$ $h, \mu, v$}. (**b**) An example of using oriented object representation.

### 3.2. Oriented Object Representation

We propose a simple representation scheme for arbitrary oriented object with a external bounding box and two gliding variables on adjacent sides of this bounding box. An intuitive illustration of our proposed representation manner is depicted in Figure 2a. The pink horizontal bounding box is the external rectangle of the green oriented bounding box $\Gamma$ with its vertexes being marked with orange dots. For the given box $\Gamma$, we use six variables

$\Gamma = \{x, y, w, h, \mu, \nu\}$ to represent a predicted oriented object computed by Equation (1). With these six variables, we can directly compute the coordinates of vertex $v_1$ and $v_2$ and refer the the coordinates of vertex $v_3$ and $v_4$ with the geometrical constraint of symmetry. Here, $\alpha$ denotes the offset of vertex $v_1'$ with respect to $v_1$, the extra variable $\mu$ is defined as the ratio between the offset $\alpha$ and the width $w$, so $\mu$ is limited to [0, 1]. According to the symmetry, $-\mu$ denotes the ratio between the offset $-\alpha$ of vertex $v_3'$ relative to vertex $v_3$ and the width $w$. Similarly, $\beta$ is the offset of vertex $v_2'$ with respect to vertex $v_2$, the extra variable $\nu$ is defined as the ratio between the offset $\beta$ and the height $h$, so $\nu$ is also limited to [0, 1]. $-\nu$ stands for the ratio between the offset $-\beta$ of vertex $v_4'$ relative to vertex $v_4$ and the height $h$. Thus, the coordinate positions of four vertexes of this oriented rectangular box can be described as follows:

$$\begin{cases} v_1 = (x - w/2, \ y - h/2) + (\mu \cdot w, \ 0) \\ v_2 = (x + w/2, \ y - h/2) + (0, \ \nu \cdot h) \\ v_3 = (x + w/2, \ y + h/2) + (-\mu \cdot w, \ 0) \\ v_4 = (x - w/2, \ y + h/2) + (0, \ -\nu \cdot h) \end{cases} \tag{2}$$

In the representation manner, the oriented object can be obtained by predicting 4 parameters $\{x, y, w, h\}$ for its external rectangle box and inferring 2 parameters $\{\mu, \nu\}$ for the offset. The advantage of this manner of representation lies in that it can reduce the ambiguity of angle in different definitions of rotation box, for example, different versions of OpenCV [6] and different definition modes based on longer side [5,13] have various intervals of definition and sign symbols for orientation angle. An example of using this method is depicted in Figure 2b. The definition of loss function of oriented object is nearly the same as that in [10,13,27].

Comparison with representations by gliding. The original object representation based on gliding vertex [10] uses 4 gliding offset on their corresponding sides of the HBB to represent a rotated object and an obliquity factor $r$ to characterize the tilt degree of the rotated object. Thus, in the regression layer of the detection head, each object regresses 9 parameters, including a 4-parameter HBB, 4 offsets characterizing a rotated quadrangle, and one obliquity factor $r$ explaining whether this target is oriented or not. This method employs horizontal RoIs to perform classification and rotated bounding box to execute regression, which will bring severe misalignment between oriented object and horizontal region feature and finally leads to the inconsistency between the high classification score and low localization accuracy. In comparison, our proposed method only regresses 2 gliding offset variables on adjacent sides of the bounding box, more importantly, we employ oriented anchors to perform classification and regression, which ensures the alignment between oriented object and oriented RoI feature.

### *3.3. Rotation-Equivariant Backbone Network*

The rotation-equivariant property is expressed as: the extracted feature from the transformed image with a rotation transformation $T_r$ being applied to the input image is the same with the one by rotating feature directly extracted from the input image. Let $I$ and $R_e$ be the input image and the rotation-equivariant extracting network, it can be expressed as:

$$R_e(T_r(I)) = T_r(R_e(I)) \tag{3}$$

Specifically, The Equation (3) shows the operation order of this feature extracting network $R_e$ and the transformation function $T_r$ is commutative.

The vast majority of object detection methods based on CNN usually adopt ResNet [39] with FPN [40] as the backbone to extract deep hierarchy of features with enriched semantic information. The convolutional weight sharing of regular CNN network ensures the inference to be translation-equivariant, which means that a translated natural input image only results in a corresponding translation of the feature map. However, objects in unconstrained

remote sensing images or aerial images are often distributed with arbitrary directions and the regular CNNs are not rotation-equivariant in the two dimensions of space and orientation. Therefore, we re-implement the Euclidean group E(2)-equivariant networks [14] into all layers (convolution, pooling, non-linearities, normalization) of the classical backbone of ResNet [39] and FPN [40] to generate rotation-equivariant ResNet (ReResNet) [15] and rotation-equivariant FPN (ReFPN) to extract the rotation-equivariant feature.

The Euclidean group E(2) is the isometry group of the plane $\mathbb{R}^2$, including translations, rotations and reflections [14]. The Euclidean group models the variation in characteristic patterns in images which occurs at arbitrary positions and in arbitrary orientations. This is especially appropriate for remote sensing images without a preferred global orientation. Specifically, all layers of the regular backbone are incorporated into the rotation-equivariant network based on e2cnn. To save the computational resources, ReResNet with ReFPN are equivariant to all translations in the image space but only $N$ discrete cyclic rotations by angles multiple of $2\pi/N$. The rotation-equivariant backbone takes a given original image as input, and outputs the feature map $f$ with the shape of ($W$, $H$, $N$, $K$), here, $W$, $H$, $N$ and $K$ stand for width, height, discrete orientation channels and position-sensitive bins of the feature map, respectively. One of the most important characteristics of this feature map is that it explicitly encodes the orientation information with $N$ discrete channels, here, each discrete orientation channel corresponds to an element. Compared with regular backbones, this backbone has three main advantages [15]: more sharing of weight parameters, not only including translation, but also rotation; enriched orientation information with multiple discrete channels ($N$ discrete orientations) and smaller model size owing to sharing of rotation weight.

### 3.4. Rotation-Invariant Feature with RiRoI Align

In general, the predicted oriented proposal is a parallelogram, whereas compared with that, the RoI projected to feature map calls for a oriented rectangular box. We enlarge the shorter diagonal of the parallelogram to have the same length with the longer diagonal while maintaining the center position fixed, and based on this, we adjust a parallelogram to a rectangular box. In this simple way, we obtain the rotated rectangular box ($x$, $y$, $w$, $h$, $\theta$). $\theta$ gives the orientation of this rectangular box and is defined by the intersection angle between the positive $X$-axis and the longer side $w$. $\theta$ is limited to $[0, \pi]$.

After obtaining oriented proposals, we are still unable to extract completely rotation-invariant features by directly applying the RRoI Align as in [5,13]. The reason lies in that RRoI Alignment can only align region feature projected to feature map according to predicted bounding box of RoI, whereas the feature map does not explicitly encode the orientation information of oriented object and cannot be aligned with special orientation module. Here, we employ the RiRoI Align module to obtain rotation-invariant feature in both dimensions.

As shown in the overall architecture of Figure 1, RiRoI Align consists of two modules: (1) Spatial alignment module. Given an RRoI $R(x, y, w, h, \theta)$ and the input feature $F$ with the size ($W$, $H$, $C$ ($C = N \cdot K$)), firstly, spatial alignment projects $R$ to feature map $F$ with a stride variable $s$ to obtain mult–scale rotated RoI $R_s(x_s, y_s, w_s, h_s, \theta)$. Secondly, it divides equally the region of $R_s$ into $L \times L$ subregions and generates the fixed-size feature $F_s$ with the size ($L \times L \times C$). For each subregion with index ($i$, $j$) ($0 \leq i, j < L$) in the channel $c$ ($0 \leq c < C$), $F_s$ is calculated:

$$F_s(i, j) = \sum_{(x,y) \in subregion(i,j)} F_{i,j,c}(T(x, y, \theta))/n \tag{4}$$

here, $n$ is the number of samples localized in the subregion. The subregion ($i$, $j$) stands for the coordinate set in the area with index ($i$, $j$). $T(\cdot)$ denotes the rotation transformation as in [5]. In this manner, we generate rotation-invariant region features $F_s$ in the spatial di-

mension. (2) Orientation alignment module. For the input region features $F_s$, the alignment in the orientation dimension is depicted as follows:

$$F_o = Int(Cdc(F_s, \lambda), \theta)), \quad \lambda = \lfloor N\theta/2\pi \rfloor \tag{5}$$

where the index $\lambda$ is an integer and is rounded down, denoting which discrete channel the oriented angle $\theta$ of an RRoI localized in. *Cdc* stands for the converting circularly discrete channel to make sure that the corresponding channel of the index $\lambda$ is the first orientation channel, the red channel in Figure 1, *Int* expresses interpolating the orientation information with the nearest *m* orientation channels. In this way, we gain features with rotation-invariant property in position and orientation.

## 4. Experiments and Discussion

To evaluate our proposed method, we perform massive experiments on the most widely used open benchmarks in remote sensing datasets: DOTA [1] and HRSC2016 [17].

### 4.1. Datasets

DOTA is a large-scale and more challenging benchmark image dataset. It is dedicated to the object detection with four-vertex positions of a quadrilateral box being annotated. It released the basic DOTA-v1.0 version and the updated DOTA-v1.5 version. DOTA-v1.0 collects 2806 remote sensing images collected from GaoFen-2 and JiLin-1 satellites and Google Earth. The image resolution in size is from 800 × 800 pixels to 4000 × 4000 pixels, the maximum resolution in space is 0.1 m, while the minimum is 4.5 m. Overall, 188, 282 object instances are covered by the following 15 common categories. According to the heights of OBBs, object instances are divided into three splits. (i) The small split in the range of 10 to 50 pixels includes object instances of all Small Vehicles (SV), Ships (SH) and partial Large Vehicles (LV); (ii) The large split with height above 300 pixels consists of all Helicopters (HC) and partial Storage tanks (ST); (iii) The medial split within the limit of 50 to 300 pixels includes remaining all Bridges (BR), Baseball diamonds (BD), Basketball courts (BC), Ground track fields (GTF), Harbors (HA), Planes (PL), Roundabouts (RA), Swimming pools (SP), Soccer-ball fields (SBF), Tennis courts (TC) and partial LVs, STs [41]. The updated version 1.5 shares the same images as version 1.0, however, a new category of Container Crane (CC) and extensive missed instances of less than 10 pixels are additionally annotated. It contains 402,089 object instances in total. Compared with DOTA-v1.0, DOTA-v1.5 is more challenging but stable during training. Some visualized examples and detection results of DOTA-v1.5 dataset are depicted in Figure 3.

The proportions for the training set (1411 images), validating set (458 images) and testing set (937 images) are 1/2, 1/6, and 1/3, respectively. The training and validation sets are incorporated together to train a model and the test set is applied to refer this model. The official metric to evaluate detection performance is mean average precision (mAP). The detection accuracy is obtained by submitting testing results to DOTA's evaluation server at https://captain-whu.github.io/DOTA/evaluation.html (accessed on 20 October 2022).

HRSC2016 is dedicated to high-resolution, rotated ship detection in remote sensing images, which contains 1061 images annotated with rotated rectangles. The ships are composed of 25 different types at deep sea or inshore harbor. The size of the images ranges from 300 × 300 to 900 × 1500. The range of the spatial resolution is from 0.4 m to 2 m. Both the training set (436 images) and validation set (181 images) are used for training a model and the test set (444 images) for testing the model. Random flipping is applied during training. The standard evaluation protocol in terms of mAP is also used.

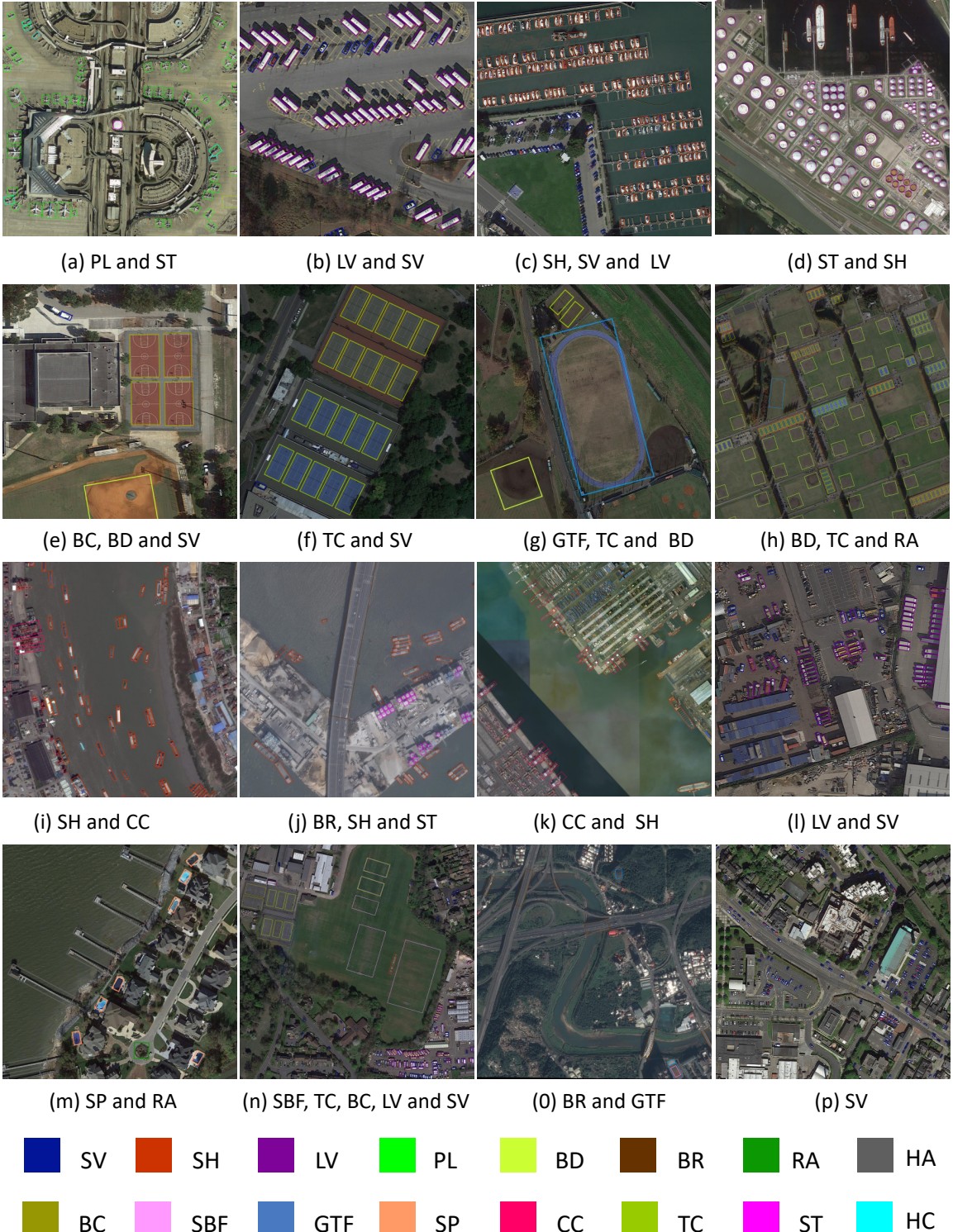

**Figure 3.** Visualization of experimental results with our RiDOP on the task of oriented object detection in DOTA-v1.5. The confidence threshold is set to 0.3 when visualizing these results. One object category is indicated with one color. The first object category of the experimental results in each sub-figure stands for the largest number of, most important detection category. (Best viewed in color and zoomed-in view.)

### 4.2. Evaluation Metrics

To evaluate the performance of object detectors, the most convincing metric is mAP, which is the combination with the Precison (P), the Recall (R), P–R curve and Mean Precision (AP). The precision P is defined as the true positive samples relative to all positive ones predicted by the detector, the recall R reflects true positive samples relative to all positive ones really existed in images (judged or labeled by human). According to the values of P and R, the AP of single-class object can be obtained by integration in the smooth P-R curve with P values, R value as the vertical axis, horizontal axis, respectively.

$$AP = \int_0^1 p_{smooth}(r)dr, \quad p_{smooth}(r) = \max_{r' >= r} p(r') \tag{6}$$

where $p_{smooth}(r)$ denotes the max $p$ value located in the right of position $r$ in the P–R curve. The mAP is the mean of all APs in multi-class evaluation.

### 4.3. Implementation Details

We use two NVIDIA Quadro RTX 6000 with the batch size of 2 in the training. The testing was executed on a single Quadro RTX 6000. The experiments were conducted on the open MMDetection [42] platform. We only adopted ReResNet50 [15] with ReFPN [40] as the backbone of the baseline method to simplify comparison and reduce computational budget. For Oriented RPN, an anchor is recognized as a positive sample with an Intersection over Union (IoU) overlap higher than 0.7 and negative sample with IoU lower than 0.3. During the training, we sample 512 RoIs with a 1:3 positive-to-negative ratio and set 15 horizontal anchors on each location of per level of the five pyramid layers. In order to reduce the redundancy, we only leave 2000 RoIs in each pyramid level before Non-Maximum Suppression (NMS) operation and choose top-2000 RoIs in total based on their classification scores after NMS. We adopt the SGD method to optimize the training network, set the weight decay to $1 \times 10^{-4}$ and the momentum to 0.9. The initial learning rate is set to 0.005 (2 GPUs). We train the network with 12 epochs and divide by 10 at epoch 8 and 11. The poly NMS threshold is set to 0.1 when merging image patches.

On the DOTA datasets, for the single-scale experiments, we crop the original large-size images with different resolutions into patches with the fixed size of 1024 × 1024. The overlap between two adjacent patches is 200 pixels. The numbers of total chip images for training and testing are 21,046 and 10,833, respectively. To compare fairly with other algorithms, random horizontal flipping and random rotation as data augmentation are adopted to avoid over-fitting in the training phase and not other complicated strategies. In the multi-scale experiments, we first resize the original images at three different scales (0.5, 1.0 and 1.5) and then crop them into 1024 × 1024 patches with the stride of 512. The numbers of total chip images for training and testing increase sharply to 143,706 and 74,058, respectively.

For the HRSC2016 dataset, we maintain the aspect ratios of images unchanged. We resize the shorter sides to 800 pixels while the longer sides to no more than 1333 pixels. We train the network with 36 epochs and divide by 10 at epoch 24 and 33. The backbone, the positive to negative ratio of RoIs, the optimization parameters and the initial learning rate, etc., are the same as the ones on DOTA.

*4.4. Peer Comparison*

Results on DOTA-v1.0. We compare our RiDOP method with other 15 state-of-the-art ones—7 one-stage methods and 8 two-stage algorithms. The detailed quantitative comparison results are shown in Table 1. The meanings of backbone networks are as follows: R50 and R101 stand for ResNet-50 and ResNet-101, respectively, Darknet-19 and H-104 denotes 19-layer darknet and 104-layer hourglass network [43], respectively, ReR50-ReFPN means rotation-equivariant ResNet-50 backbone with rotation-equivariant FPN network, DLA-34 is 34-layer network combined resnet with densenet, HRGANet depicts gated aggregation network. The one-stage methods are also expressed as anchor-free methods, which do not generate anchors according to the ground truth and do evaluate whether the predicted RoI belongs to an object class or not. The characteristic is a double-edged sword between fast running speed and relatively low accuracy, such as state-of-the-art RSDet [44] and S$^2$A-Det [11], which have gained 70.80% mAP in the single-scale testing and 79.74% mAP in the multi-scale experiments, respectively. Owing to the complexity in appearance, scale and orientation, two-stage algorithms with well-designed modules are usually executed on the DOTA dataset. To better describe oriented objects, such as CenterMap [45] by predicting the probability to regress OBBs, Gliding Vertex [10] by sliding to represent OBBs, have achieved 71.74% mAP and 75.02% mAP, respectively. To overcome the delicate feature following the variety of orientation, ReDet [15] with rotation-equivariant network has got 76.25% mAP in the single-scale experiments and 80.10% mAP in the multi-scale experiments, respectively.

Without tricks, in the single-scale experiments, our RiDOP obtains 76.50% mAP with ReR50-ReFPN backbone and outperforms all single-scale methods and most multi-scale ones. In addition, with multi-scale strategies, our RiDOP achieves 80.26% mAP in the whole dataset, which outperforms all state-of-the-art methods.

Results on DOTA-v1.5. Compared with DOTA-v1.0, DOTA-v1.5 is more challenging with much less than 10 pixel object instances (most commonly, extremely small categories of small vehicle and ship), which improves the difficulty of object detection. We compare our RiDOP method with other five state-of-the-art ones in the single-scale experiments. The detailed quantitative comparison results are shown in Table 2. Here, R2CNN [46], TPR-R2CNN [47], and FasterRCNN-O [27] are the varieties of the classical two-stage detection method of RCNN. TPR-R2CNN [47] is the improved version of R2CNN [46] with two detection heads, which improves the detection accuracy from 54.52% mAP to 58.42% mAP; in addition, FasterRCNN-O [27] outperform TPR-R2CNN by 3.58% in precision. RetinaNet-O [18] is the one-stage method with better class balance between hard samples and easy samples, which has achieved 59.19% mAP. ReDet [15] is dedicated to oriented object detection with rotation-invariant feature, which achieved 66.86% mAP with an obvious improvement. Without tricks, our RiDOP method achieves 67.99% mAP, which outperforms all the state-of-the-art methods. Compared with previous best results in ReDet [15], our RiDOP gains the improvement of 1.13% mAP (from 66.86% mAP to 67.99% mAP). In contrast with Table 1, our RiDOP gains the higher increment (from 0.25% mAP to 1.13% mAP) in the more challenging dataset, which demonstrates the better robustness of our RiDOP method. Some qualitative visualization results are shown in Figure 3.

**Table 1.** Quantitative comparison with state-of-the-art algorithms on the DOTA-v1.0. MS denotes the methods trained and test using multi-scale images. '✓' represents using multi-scale images, '-' denotes the opposite. The metric for each category is AP and for overall ones, it is mAP.

| Method | Backbone | MS | PL | BD | BR | GTF | SV | LV | SH | TC | BC | ST | SBF | RA | HA | SP | HC | mAP |
|---|---|---|---|---|---|---|---|---|---|---|---|---|---|---|---|---|---|---|
| **One-stage:** | | | | | | | | | | | | | | | | | | |
| YOLOV2 [26] | Darknet-19 | - | 52.75 | 24.24 | 10.60 | 35.50 | 14.36 | 2.41 | 7.37 | 51.79 | 43.98 | 31.35 | 22.30 | 36.68 | 14.61 | 22.55 | 11.89 | 25.49 |
| PIoU [32] | DLA-34 | - | 80.90 | 69.70 | 24.10 | 60.20 | 38.30 | 64.40 | 64.80 | 90.90 | 77.20 | 70.40 | 46.50 | 37.10 | 57.10 | 61.90 | 64.00 | 60.50 |
| RSDet [44] | R50-FPN | - | 89.30 | 82.70 | 47.70 | 63.90 | 66.80 | 62.00 | 67.30 | 90.80 | 85.30 | 82.40 | 62.30 | 62.40 | 65.70 | 68.60 | 64.60 | 70.80 |
| O$^2$-DNet*[48] | H104 | - | 89.30 | 83.30 | 50.10 | 72.10 | 71.10 | 75.60 | 78.70 | 90.90 | 79.90 | 82.90 | 60.20 | 60.00 | 64.60 | 68.90 | 65.70 | 72.80 |
| DRN [29] | H104 | ✓ | 89.71 | 82.34 | 47.22 | 64.10 | 76.22 | 74.43 | 85.84 | 90.57 | 86.18 | 84.89 | 57.65 | 61.93 | 69.30 | 69.63 | 58.48 | 73.23 |
| OSSDet [49] | R101-FPN | ✓ | 89.49 | 81.10 | 51.23 | 71.30 | 76.80 | 76.97 | 87.27 | 90.79 | 83.43 | 84.71 | 60.55 | 64.92 | 71.21 | 70.44 | 66.00 | 75.08 |
| S$^2$A-Net [11] | R50-FPN | ✓ | 88.89 | 83.60 | 57.74 | 81.95 | 79.94 | 83.19 | 89.11 | 90.78 | 84.87 | 87.81 | 70.30 | 68.25 | 78.30 | 77.01 | 69.58 | 79.42 |
| **Two-stage:** | | | | | | | | | | | | | | | | | | |
| RADet [50] | R101-FPN | - | 79.45 | 76.99 | 48.05 | 65.83 | 65.46 | 74.40 | 68.86 | 89.70 | 78.14 | 74.97 | 49.92 | 64.63 | 66.14 | 71.58 | 62.16 | 69.09 |
| CenterMap [45] | R50-FPN | - | 88.88 | 81.24 | 53.15 | 60.65 | 78.62 | 66.55 | 78.10 | 88.83 | 77.80 | 83.61 | 49.36 | 66.19 | 72.10 | 72.36 | 58.70 | 71.74 |
| ReDet [15] | ReR50-ReFPN | - | 88.79 | 82.64 | 53.97 | 74.00 | 78.13 | 84.06 | 88.04 | 90.89 | 87.78 | 85.75 | 61.76 | 60.39 | 75.96 | 68.07 | 63.59 | 76.25 |
| RoI Trans. [5] | R101-FPN | ✓ | 88.64 | 78.52 | 43.44 | 75.92 | 68.81 | 73.68 | 83.59 | 90.74 | 77.27 | 81.46 | 58.39 | 53.54 | 62.83 | 58.93 | 47.67 | 69.56 |
| CenterRot. [51] | R101-FPN | ✓ | 89.74 | 83.57 | 49.53 | 66.45 | 77.07 | 80.57 | 86.97 | 90.75 | 81.50 | 84.05 | 54.14 | 64.14 | 74.22 | 72.77 | 54.56 | 74.00 |
| Gliding Vertex [10] | R101-FPN | ✓ | 89.64 | 85.00 | 52.26 | 77.34 | 73.01 | 73.14 | 86.82 | 90.74 | 79.02 | 86.81 | 59.55 | 70.91 | 72.94 | 70.86 | 57.32 | 75.02 |
| BBAVectors[52] | R101 | ✓ | 88.63 | 84.06 | 52.13 | 69.56 | 78.26 | 80.40 | 88.06 | 90.87 | 87.23 | 86.39 | 56.11 | 65.62 | 67.10 | 72.08 | 63.96 | 75.36 |
| RIE [41] | HRGANet-W48 | ✓ | 89.85 | 85.68 | 58.81 | 70.56 | 76.66 | 82.47 | 88.09 | 90.56 | 80.89 | 82.27 | 60.46 | 63.67 | 76.63 | 71.56 | 60.89 | 75.94 |
| ReDet [15] | Re50-ReFPN | ✓ | 88.81 | 82.48 | 60.83 | 80.82 | 78.34 | 86.06 | 88.31 | 90.87 | 88.77 | 87.03 | 68.65 | 66.90 | 79.26 | 79.71 | 74.67 | 80.10 |
| **Our (two-stage):** | | | | | | | | | | | | | | | | | | |
| RiDOP | Re50-ReFPN | - | 89.37 | 82.54 | 55.64 | 75.25 | 73.38 | 83.65 | 88.30 | 90.82 | 86.36 | 84.98 | 60.55 | 62.19 | 75.59 | 72.12 | 66.76 | 76.50 |
| RiDOP | Re50-ReFPN | ✓ | 89.33 | 84.99 | 60.70 | 80.41 | 79.40 | 84.38 | 88.69 | 90.89 | 88.70 | 87.74 | 69.81 | 68.12 | 78.35 | 78.59 | 73.82 | 80.26 |

**Table 2.** Quantitative comparison with state-of-the-art methods on the DOTA-v1.5. The metric for each category is AP and for overall ones, it is mAP.

| Method | PL | BD | BR | GTF | SV | LV | SH | TC | BC | ST | SBF | RA | HA | SP | HC | CC | mAP |
|---|---|---|---|---|---|---|---|---|---|---|---|---|---|---|---|---|---|
| R2CNN [46] | 87.74 | 60.84 | 43.16 | 64.87 | 45.75 | 48.39 | 69.85 | 56.19 | 54.78 | 71.06 | 50.59 | 64.03 | 54.00 | 53.43 | 47.65 | 0.0 | 54.52 |
| TPR-R2CNN [47] | 88.79 | 60.30 | 49.65 | 65.24 | 48.19 | 55.96 | 75.37 | 68.56 | 56.15 | 71.03 | 50.68 | 66.48 | 64.25 | 57.30 | 56.61 | 0.26 | 58.42 |
| RetinaNet-O [18] | 71.43 | 77.64 | 42.12 | 64.65 | 44.53 | 56.79 | 73.31 | 90.84 | 76.02 | 59.96 | 46.95 | 69.24 | 59.65 | 64.52 | 48.06 | 0.83 | 59.16 |
| FasterRCNN-O [27] | 71.89 | 74.47 | 44.45 | 59.87 | 51.28 | 68.98 | 79.37 | 90.78 | 77.38 | 67.50 | 47.75 | 69.72 | 61.22 | 65.28 | 60.47 | 1.54 | 62.00 |
| ReDet [15] | 79.20 | 82.81 | 51.92 | 71.41 | 52.38 | 75.73 | 80.92 | 90.83 | 75.81 | 68.64 | 49.29 | 72.03 | 73.36 | 70.55 | 63.33 | 11.53 | 66.86 |
| RiDOP (Our) | 79.98 | 83.09 | 55.21 | 65.82 | 52.39 | 76.34 | 87.99 | 90.90 | 81.52 | 67.34 | 49.88 | 70.85 | 73.09 | 73.08 | 68.54 | 11.79 | 67.99 |

Results on HRSC2016. The HRSC2016 dataset contains a large number of small ship instances with rich variations in rotation, scale, position, shape and appearance , etc. It is still more challenging for single-category small object detection. To demonstrate the superiority, 10 state-of-the-art methods are used for fair comparison and their results are depicted in Table 3 using the mAP value under PASCAL VOC 2007 and VOC 2012 metrics. Our method achieves the best performance with 90.54% mAP under the methric of VOC2007 and 98.47% mAP under the metric of VOC2012 using ResNet-50 as the backbone, respectively. Qualitative visualization results are shown in Figure 4.

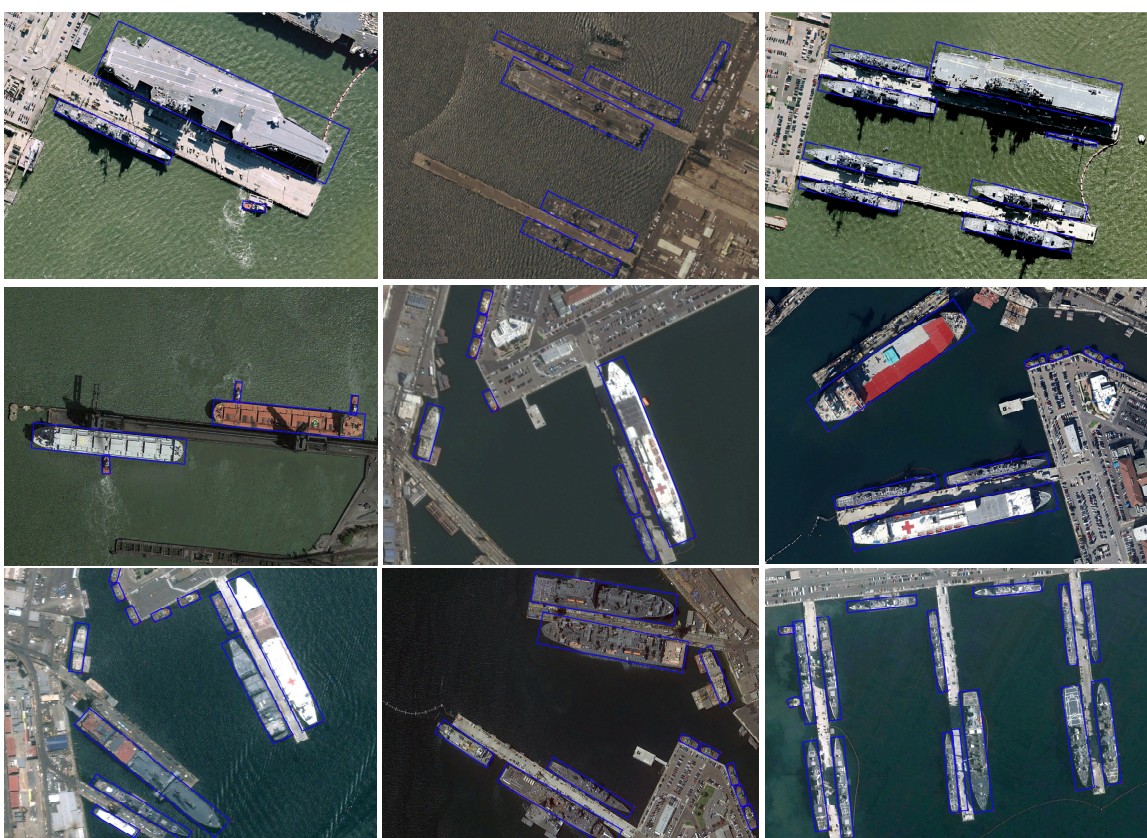

**Figure 4.** Visualization of experimental results with our RiDOP algorithm on the task of oriented object detection in the HRSC2016 dataset. The confidence threshold is set to 0.3 when visualizing these results. (Best viewed in color and zoomed-in view.)

Model size and speed. For fair comparison, we evaluate the model size versus detection accuracy of different algorithms on DOTA-v1.5 under the same setting. The comparison result with five state-of-the-art methods is depicted in Table 4. All methods use ResNet50 or its variation as the backbone. Compared with previous best results, our RiDOP not only gains the improvement of 1.13% mAP (from 66.86% to 67.99%), but also achieves smaller model size by a large margin (121 Mb vs. 71 Mb). For reference speed, the hard-

ware platform of testing is a single NVIDIA Quadro RTX 6000 GPU with batch size of 2. During testing, we run at the speed of 10.4 FPS (Frames Per Second) with the image size of 1024 × 1024. Our method achieves better accuracy-versus-speed trade-off.

**Table 3.** Comparison with state-of-the-art algorithms on the HRSC2016.

| Methods | Backbone | mAP (VOC 07) | mAP (VOC 12) |
|---------|----------|--------------|--------------|
| RoI Trans. [5] | R101-FPN | 86.20 | – |
| Gliding Vertex. [10] | R101-FPN | 88.20 | – |
| PIoU [32] | DLA-34 | 89.20 | – |
| DRN [29] | H-34 | – | 92.70 |
| R$^3$Det [12] | R101-FPN | 89.26 | – |
| CenterMap [45] | R101-FPN | – | 92.80 |
| S$^2$A-Net [11] | R50-FPN | 90.17 | 95.01 |
| OR-CNN [13] | R50-FPN | 90.40 | 96.50 |
| CenterRot [51] | R50-FPN | 90.20 | 96.59 |
| ReDet [15] | Re50-ReFPN | 90.46 | 97.63 |
| RiDOP (Our) | Re50-ReFPN | 90.54 | 98.47 |

**Table 4.** The detection accuracy (mAP) versus model size on DOTA-v1.5. We compare 5 state-of-the-art methods, including Faster RCNN-O [27], RetinaNet-O [18], RoI Transformer [5], Hybrid Task Cascade (HTC) [53,54] and ReDet [15]. For fair comparison, the backbones of all methods adopt uniformly ResNet50 or the variant. Our RiDOP not only outperforms all methods in the detection accuracy, but also achieves smallest model size by a large margin (121 Mb vs. 71 Mb). Moreover, our method runs with comparable speed (10.4 FPS) on one RTX 6000 GPU.

| Methods | Framework | Backbone | mAP (%) | Param (Mb) | Platform | FPS |
|---------|-----------|----------|---------|------------|----------|-----|
| Faster RCNN-O [27] | Two-stage | R50 | 62.00 | 158 | Tesla V100 | 14.1 |
| RetinaNet-O [18] | One-stage | R50 | 62.67 | 140 | Tesla V100 | 12.1 |
| RoI Trans. [5] | Two-stage | R50 | – | 273 | TITAN X | 5.9 |
| HTC [53] | Two-stage | R50 | 63.40 | 295 | Tesla V100 | 7.9 |
| ReDet [15] | Two-stage | ReR50 | 66.86 | 121 | RTX 6000 | 9.9 |
| RiDOP (Our) | Two-stage | ReR50 | 67.99 | 71 | RTX 6000 | 10.4 |

*4.5. Discussion*

By quantitative numeric comparison and qualitative visualized analysis on several challenging remote sensing image datasets, the effectiveness and superiority of our method is verified. Our RiDOP can offer the state-of-the-art performance to locate arbitrary-oriented objects with rich variations in appearance, scales, position, shape, orientation, etc. In addition, it also can reduce the number of parameters by a huge margin.

In Table 2, the mAP of all 16 categories of our method reaches 67.99%, in contrast, the AP of container crane category is just 11.97% with a drastic drop in accuracy. The reason lies in the extreme category imbalance encountered when training the detection model. According to the statistics, the number of object instances of container crane is 237, accounting for less than 0.06 percent in the overall 402,089 instances. Conversely, the share of instances of small vehicle reaches 60 percent. The number ratio of the maximal instances to the minimal ones is up to 100. The severe category imbalance leads to that the parameters of the model are optimized in the direction of being beneficial to categories with large numbers of samples. However, other small categories, such as helicopter, basketball court and roundabout are about 4 to 5 times as many as container crane, and obtain unexpectedly more than 68% mAP with a sharp improvement. This phenomenon shows that the category imbalance is just one of reasons leading to the drastic drop of the performance. In the

following research, on the one hand, we will make use of some strategies, for instance, data augmentation, collecting more images and giving sensitive weights, to reduce the effect of the category imbalance. On the other hand, we will pay more attention to the generalization to improve the robustness.

Furthermore, when applying zoomed-in view in Figure 3h, the detection of some small vehicles with highly similar appearance in the left bottom of the image are missed, other scenes also exist the same phenomenon. The objects with highly similar appearances cannot be given consistent results with the same detection network. This is a significant research topic of the complicated issues about the interpretability of the feature, the visualization of sematic information and how to evaluate the difference between human and computer, which will be addressed in our future research.

## 5. Conclusions

The oriented object detection in remote sensing images is of high interest for several applications including traffic monitoring, urban planning, disaster management, searching and rescuing. In this paper, we propose a novel object detection method which combines the simple oriented object representation manner with rotation-invariant feature extracting architecture to detect densely displayed oriented objects with arbitrary orientations in remote sensing images. It consists of two key components: (i) a simple yet effective oriented RPN to generate high-quality oriented proposals and use these proposals to represent oriented objects by sliding two vertexes only on adjacent sides of the external rectangle, which is a lightweight network without complicated parameters or modules to learn; (ii) the rotation-equivariant backbone to obtain the feature map with explicit orientation channel information being modeled and the RiRoI Alignment module to obtain completely rotation-invariant features in both spatial and orientation dimensions. Without tricks, our RiDOP achieved the state-of-the art accuracy with 67.99% mAP (increased by 1.13% mAP) on the more challenging DOTA-v1.5 and 98.47% mAP (increased by 0.84% mAP) on HRSC2016, while reducing the number of parameters by 40% (121 Mb versus 71 Mb) in comparison with the previous best result. Extensive experiments conducted on the challenging datasets demonstrate the effectiveness and superiority of our proposed method.

**Author Contributions:** Conceptualization, C.W.; Data curation, W.N.; Formal analysis, Y.Q. and J.W.; Investigation, Q.L. and K.C.; Methodology, C.W.; Project administration, W.N.; Resources, H.Z.; Validation, C.W.; Writing—original draft, C.W.; Writing—review and editing, H.B. All authors have read and agreed to the published version of the manuscript.

**Funding:** This research was supported in part by the National Natural Science Foundation of China under Grant 42101344.

**Data Availability Statement:** Two open-source datasets associated with the research are available online. The DOTA dataset is available at https://captain-whu.github.io/DOTA.html (accessed on 20 October 2022). The HRSC2016 dataset can be downloaded from Kaggle website and is available at https://www.kaggle.com/guofeng/hrsc2016 (accessed on 20 October 2022).

**Acknowledgments:** We acknowledge the use of two open-source benchmark datasets of DOTA and HRSC2016. We also feel grateful for the contributions and comments of the editors and anonymous peer reviewers.

**Conflicts of Interest:** The authors declare no conflict of interest.

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
