# Peer review of "RiDOP: A Rotation-Invariant Detector with Simple Oriented Proposals in Remote Sensing Images"

_remotesensing, doi:10.3390/rs15030594_

Round 1

Reviewer 1 Report

First of all, I would like to thank you for the article submission.

The primary aim of the paper was to develop a NN architecture to detect objects invariant of rotation with simple oriented proposals for remote sensing images with state-of-the-art performance.

There are some comments and suggestions that I want to express:

The text should be edited - There are some English language errors throughout the article that need correction. Also, there are many words and sentences that are subjective, exaggerated, vague, and informal which need to be rectified. (eg. "very gratifying", "bell and whistles", "excellent", "most outstanding", etc.).

Peer comparison - In the peer comparison chapter, you compare the difference of mAP between consecutive algorithms from the table which is confusing ( there are mentions like "more growth" and "surpass" that are not accurate).

Just mentioning the higher mAP of your method should be enough for this chapter.

In the text you mention that your model mAP for single-scale images is 76.5 % but in the table is shown as trained with multi-scale images. Which one is correct?

Why for Dota -v1.5 you only analyze single-scale data? I noticed that the data shown for ReDet[2] in table 2 for multi-scale has a significantly higher mAP. 

Final questions and suggestions:

What is the number of parameters of this architecture? It would be relevant to analyze this and the speed of your proposed model in comparison with other state-of-the-art models like the ones other articles on this subject have done. 

Why do you think your model showed better results than the others analyzed?

Reviewer 2 Report

Overall, the paper is good. Unlike some methods that show their novelties using complex steps, this paper presents a simple but effective method. I think simpler is better in current deep learning time. 

Only some minor comments should be addressed. 

(1) backbobe network is confusing. 

(2) do not use double lines in tables. 

(3) the conclusion is too short that the main contributions and the proposed method are not well summarize. Please rewrite the conclusion. 

Round 2

Reviewer 1 Report

The text should be extensively edited for the English language and style.
